# The Development of Honey Recognition Models Based on the Association between ATR-IR Spectroscopy and Advanced Statistical Tools

**DOI:** 10.3390/ijms23179977

**Published:** 2022-09-01

**Authors:** Maria David, Ariana Raluca Hategan, Camelia Berghian-Grosan, Dana Alina Magdas

**Affiliations:** National Institute for Research and Development of Isotopic and Molecular Technologies, 67-103 Donat Street, 400293 Cluj-Napoca, Romania

**Keywords:** honey authentication, ATR-IR spectroscopy, PLS-DA, preprocessing

## Abstract

The newly developed prediction models, having the aim to classify Romanian honey samples by associating ATR-FTIR spectral data and the statistical method, PLS-DA, led to reliable differentiations among the samples, in terms of botanical and geographical origin and harvesting year. Based on this approach, 105 out of 109 honey samples were correctly attributed, leading to true positive rates of 95% and 97% accuracy for the harvesting differentiation model. For the botanical origin classification, 83% of the investigated samples were correctly predicted, when four honey varieties were simultaneously discriminated. The geographical assessment was achieved in a percentage of 91% for the Transylvanian samples and 85% of those produced in other regions, with overall accuracy of 88% in the cross-validation procedure. The signals, based on which the best classification models were achieved, allowed the identification of the most significant compounds for each performed discrimination.

## 1. Introduction

Honey is a natural sweet substance made by Apis mellifera bees, either from the nectar of plants, from the secretion of plants or from the excretion of insects. It has been consumed since ancient times and it is known to have high nutritional value. Honey mainly consists of carbohydrates such as fructose, glucose and sucrose, whose concentrations depend on the botanical origin. Non-volatile compounds such as sugars [1,2], amino acids [3,4], proteins, minerals [1] and phenolic compounds [5] contribute to the taste and color of the honey, while the volatile compounds confer its aroma [6].

Since honey supply is lower than the demand, it is susceptible to adulteration. Monofloral honey, being the most appreciated, is the main target of adulteration by mixing multiple types of multifloral honey. The quality, aroma and physical properties of honey within the same floral source vary due to seasonal climatic variations or the geographical location of the apiary [3]. Therefore, an important issue has been to identify pure honey and verify its authenticity, by developing analytical approaches to permit the verification of the quality specification.

The composition of honey changes over time, and the most important modifications are related to its concentrations of dextrose, levulose, maltose, sucrose, higher sugars, diastase and total acidity, which lead to darkening and loss of aroma and flavor [7]. Furthermore, during storage, the Maillard reaction and/or caramelization can occur, when the concentration of monosaccharides decreases and the levels of organic acids, 5-hydroxymehyl-2-furaldehyde (HMF) and furosine increase [8,9].

The most popular method to authenticate the botanical origin of honey is melissopalynological analysis based on microscopic identification of the pollen type [10]. However, this method is time-consuming and implies many processing steps. Therefore, there is an ongoing need to develop faster, practical methods to determine the botanical or geographical origin of the honey. In recent years, the Fourier transform infrared spectroscopy (FT-IR) has been frequently used in the food industry for the detection and quantification of adulteration [11,12], because of its fast response time and wide spectral range of analysis. Moreover, if the spectrometer is equipped with an attenuated total reflectance (ATR) sample technique, it is nondestructive and does not require any sample manipulation [6,13].

Near-infrared and mid-infrared spectroscopy in corroboration with chemometric methods were extended, developed and successfully used for the quality control and botanical origin assessment of honey samples [14,15,16,17]. Several methods have been developed using partial least squares discriminant analysis (PLS-DA) for the detection of adulterated honey, by determining the sugar syrup content [18,19] or for sugar quantification in honey, such as glucose, fructose, sucrose and maltose [10,12]. Good results in differentiating the botanical origin of 11 honey types using linear discriminant analysis (LDA), having a correct classification rate ranging from 70 to 100%, were reported in the literature by Ruoff and co-workers [10]. Moreover, a good botanical classification of honey samples from Turkey by using principal component analysis (PCA) and hierarchical cluster analysis (HCA) was obtained by Gok and co-workers [18]. The near- and mid-infrared spectra of raw honey samples were also employed for the classification of honey samples with different botanical origins, leading to accuracies greater than 96% when PLS-DA and PCA were applied [19].

For the geographical authentication, Formosa et al. used spectral transformations, variable selection and PLS-DA to successfully classify the provenance of 21 local and 49 non-local honey samples [20]. Other reported studies present the use of chemometric tools (PLS, PCA, LDA) in NIR spectroscopic studies to differentiate honey according to its geographical origins [21,22]. All these studies employed the use of spectral transformations prior to multivariate analysis.

This study proposes a new approach for the development of prediction models to classify a honey set formed by 109 Romanian samples belonging to four botanical origins that were produced in two consecutive harvesting years. The association between ATR-FTIR spectral data and a supervised statistical method, PLS-DA, led to reliable differentiations among the samples in terms of the year of harvest, botanical origin and geographical provenance. The novelty of the present work is highlighted by the construction of a highly precise discrimination model capable of distinguishing honey belonging to 2020 production from that harvested in 2021. According to our knowledge, to date, there are no reported data related to the possibility of easily differentiating the harvesting year of honey based on the association of IR spectroscopy and advanced statistical models. The practical importance of this classification is related to the detection of the possible unfair practices related to the re-utilization of unsold honey by labeling it as recently harvested. In order to maximize the prediction rate of our developed differentiation models, special emphasis was given to the applied data preprocessing step, a fact that also proved to be a very effective step in the models’ prediction performance.

## 2. Results

All honey spectra were recorded in the spectral range 550–4000 cm^−1^. For a clear understanding of the samples’ characteristics, Figure 1 contains the FTIR spectra of acacia, colza, honeydew and linden honey species (Figure 1a) and acacia samples collected in the 2020 and 2021 years, respectively (Figure 1b).

As has been demonstrated over the years in the literature [18,23,24,25,26,27], six representative regions can be emphasized in the FTIR spectra of honey (Figure 1a). Most of these bands are associated with carbohydrates and O-H stretching vibrations (both from water and carbohydrates), but the overlapping with the peaks originating from some minor compounds such as amino acids, proteins or various organic acids can be also considered in some spectral regions (i.e., spectral range from 1170 to 1540 cm^−1^, 1550 to 1750 cm^−1^ or 2800 to 3000 cm^−1^) [13,26,27,28,29].

Moreover, visible differences are observed in almost all IR regions of the investigated honey varieties (Figure 1a) and especially in three domains, namely 950 to 1170 cm^−1^, 1170 to 1540 cm^−1^ and 3000 to 3670 cm^−1^, when studying the harvesting year influences (Figure 1b). Because the honey differentiations, according to some predefined criteria, such as harvesting year and botanical and geographical origin, are mainly realized based on the minor constituents of the honey—both organic and inorganic compounds—for the classification models’ development, only the fingerprint region between 550 and 1775 cm^−1^ was further considered, in order to reduce the dimensionality of the data set.

## 3. Discussion

For the development of the most effective recognition models, the obtained experimental spectra were preprocessed using autoscale. Then, a feature selection procedure that aimed to reduce the number of input variables only to those that had the highest discrimination potential was applied. The final differentiation models were constructed by applying the PLS-DA supervised method on the input space corresponding to only those previously identified meaningful markers. A comparison of the models’ performance showed that the best recognition models were obtained when a preprocessing step was performed and when only the relevant markers were used for the development of the models.

The previously reported studies emphasized the potential of IR spectroscopy in the differentiation of the botanical and geographical origins of honey [19,20,21], but according to our knowledge, no studies related to the differentiation of honey harvesting years have been performed.

### 3.1. Harvesting Year Differentiation

For the development of the model for harvesting year differentiation, a total of 109 honey samples, 60 from 2020 and 49 from 2021, were considered. The best obtained recognition model was constructed by taking into consideration 63 markers (i.e., spectral points), as resulted from the model-based feature selection method (Figure 2). Six latent variables were used to construct the differentiation model, the selection of which was made with the aim of minimizing the cross-validation error average (Figure 3). Through this classifier, 105 out of 109 honey samples were assigned to the correct class, leading to true positive rates of 95% and 97% for the groups of 2020 and 2021, respectively. The scores, ROC and RMSECV plots associated with the prediction model are presented in Figure 4 and Figure 5.

The markers based on which the best classification was achieved were situated in the following spectral ranges: 550–700, 1050–1100 and a band around 1419 cm^−1^ (Figure 2). An important observation pertained to the fact that the number corresponding to these relevant markers was considerably lower (i.e., 63) as compared to the initial dimensionality of the entire spectra (i.e., 1246).

An important spectral range that allowed the present classification is represented by the region between 550 and 700 cm^−1^. This region is known as the “crystalline region” and contains the exocyclic deformations (CCO) [24], or it can be associated with the M-O vibrations [30], being able to offer information about the presence of some metals that are directly related to environmental conditions (i.e., soils, pollution issues).

Another spectral region that contained markers for the harvesting year differentiation is the one situated between 1050 and 1100 cm^−1^. This region is specific for the C-O stretching vibration in carbohydrates such as sucrose, glucose and fructose [31]. This range also indicates the presence of C-C stretching modes and O-H vibrations, characteristic of organic acids, carotenes and polyphenols [12,32]. Monosaccharide concentrations can be considered as an important feature for the storage time of honey, since their concentration decreases with time due to spontaneous chemical degradation, such as the non-enzymatic Maillard reaction [9].

A meaningful marker for the present classification was represented by the band around 1419 cm^−1^, attributed to a combination of O-H bending of the C-O-H group and C-H bending vibrations of alkenes [12]. The structure of some organic acids contains -CH=CH- bonds, which can contribute to this band, e.g., in the case of fumaric acid (COOH-CH=CHCOOH. The presence of fumaric acid is an indicator of the deterioration of honey because of storage and aging [7,12].

### 3.2. Botanical Differentiation

The botanical classification model was constructed based on the entire set of honey samples that corresponded to the four floral types: acacia (41 samples), colza (18), honeydew (20) and linden (30). The most accurate obtained PLS-DA classifier was built on 146 features that were selected as significant markers for the botanical differentiation of the honey samples (Figure 6). Eighteen components proved to be the optimal selection for the number of latent variables (Figure 7), leading to 83% accuracy in the cross-validation evaluation procedure. Therefore, 91 samples were correctly predicted (34 acacia, 14 colza, 11 honeydew, 24 linden samples) according to their botanical class. The score plot associated with the botanical differentiation models is illustrated in Figure 8. Moreover, the RMSECV and ROC plots are presented in Figure 9a,b.

The markers based on which the best classification was achieved were situated in the spectral ranges between 550 and 1000, and 1550 and 1700 cm^−1^.

The spectral area 550–700 cm^−1^ can be also associated with the skeletal vibrations of carbohydrates, being known as the “crystalline region”, containing the exocyclic deformations (CCO) [24]. However, few influences for botanical discrimination can be also identified in the region below 700 cm^−1^ (Figure 6).

As has been illustrated in Figure 1a, the spectral area from 700 to 900 cm^−1^ is specific to the anomeric forms of saccharides, being called the fingerprint or “anomeric region”. By analyzing the bands from this region, it was possible to identify the α and β anomers in monosaccharides or higher saccharides [24]. The peaks from 950 to 1000 cm^−1^ correspond mainly to the C-O stretching of carbohydrates, and allow differentiation according to the presence of glucose, fructose and sucrose in the honey samples. These two regions have the most significant contribution to the botanical discrimination of samples, being in accordance with the fact that various types of honey contain different amounts of glucose and fructose, respectively. This region can also indicate the presence of C-O stretching modes characteristic of polyphenols [32], which is an indicator of floral origin, as flower honey is characterized by high concentration values of glucose and fructose and low polyphenol content, whereas honeydew honey has a lower concentration of glucose and fructose and higher polyphenol content compared with other varieties of honey [33].

The spectral region around 1530–1730 cm^−1^ represents an overlapped signal involving the O-H deformation vibration of water and the C=O stretching from carbohydrates, proteins, amino acids or organic acids [18,27,29,30]. The region of 1600–1700 cm^−1^ was also involved in the botanical differentiation in previously reported studies [18,26], and its efficiency can be explained through the presence of proteins and some interactions, i.e., water–carbohydrates and water–proteins, which are directly related to the floral origin. However, in this range, the water content has to be considered as having a significant influence on the IR spectra.

All these identified meaningful spectral regions from the current work were previously used for developing models in order to differentiate the botanical origins of honey samples [10,12,18,19,26].

### 3.3. Geographical Discrimination

The PLS-DA models designed for geographical differentiation were based on 54 samples: 34 from Transylvania and 20 honey samples from other regions. Once again, the recognition model that proved to have the highest discrimination capability corresponded to a reduced input space—namely, only 28 spectral points were taken into account (Figure 10). Regarding the selection for the number of latent variables, seven components were chosen for the construction of the PLS-DA model (Figure 11). Here, 88% accuracy was achieved after conducting the cross-validation procedure, leading to the correct prediction of 91% of the Transylvanian samples and 85% of the honey samples produced in the other regions. Satisfactory separation of the samples in terms of the geographical origin is reflected through the score plot associated with the first three latent variables (Figure 12). Moreover, the RMSECV and ROC plots associated with the developed classification model are presented in Figure 13a,b.

The markers based on which the best classification was achieved are illustrated in Figure 10, being mainly situated in the region 550–700 cm^−1^, and, besides these, a few narrow bands around 710, 810, 870 and 1180 cm^−1^ also proved to have discrimination potential. The spectral region 500–700 cm^−1^ is characteristic of the skeletal vibrations of carbohydrates, but these markers can be also indicative of the soil composition [34] and can reveal the differences in the mineral content of the soil. Thus, the peaks found in this area correspond to M-O vibrations and could give information about the presence of some metals, such as aluminum, copper, lead, iron, tin [30], manganese [35], lithium [36] or strontium [37], which could be seen as possible markers of geographical origin. Manganese, lithium and strontium elements have been identified as elemental markers in various food commodities from the Transylvania area, such as milk [38], carrots [39] and cheese [40]. These elements proved to be powerful discriminators of Transylvanian products as compared to those that originated from other geographical regions.

The signals from the region between 810 and 890 cm^−1^ are characteristic of the anomeric vibrations of carbohydrates or could be due to the C-H out-of-plane deformation vibrations [28], being important for honey analysis, while the band at 1180 cm^−1^ could be associated with C-O stretching vibrations from phenols [30].

## 4. Materials and Methods

### 4.1. Sample Description

For this study, 109 authentic honey samples collected during the 2020 and 2021 harvest, directly from producers located in the most important melliferous regions of Romania, were analyzed and the obtained analytical data were further processed.

The honey floral origin, year of harvest and geographical distribution are presented in detail in Table 1. With respect to the geographical origin, it was necessary to take into account the apiculture practices in Romania. Because, for some flowers (e.g., acacia), the blooming period is very short and differs from one Romanian region to another, moving the hives in order to increase honey production is a common practice. This habit leads to the imprecise geographical origin of some honey samples. For this reason, only the samples having an exact region of collection (i.e., Transylvania) were explicitly considered to belong to a precise area.

All collected honey samples were stored in the dark, in screw-cap jars, at room temperature (18–25 °C) until the spectral acquisition. Prior to the spectral collection, the crystalized honey samples were heated overnight (37 °C) and manually stirred to ensure their homogeneity.

### 4.2. Vibrational Spectroscopy Analysis and ATR-FTIR Spectroscopy

The infrared spectra of the honey samples were recorded with a Jasco 4100 FTIR spectrometer in attenuated total reflectance (ATR) mode with a ZnSe crystal. The detection system consisted of a DTGS detector, having a spectral resolution of 4 cm^−1^. At room temperature, samples were placed on the crystal. Each spectrum was recorded in the spectral range 550–4000 cm^−1^, by averaging 16 scans. The air spectrum was taken as background, and between measurements, the ATR crystal was carefully cleaned with acetone.

### 4.3. Statistical Treatments

For the development of the honey classification models, a supervised statistical method was applied, namely partial least squares discriminant analysis (PLS-DA). PLS-DA represents a classification methodology derived from PLS regression and it involves building a regression model between the data matrix containing the input variables (i.e., measurements) and a dummy matrix that illustrates the adherence of each sample from the data set to a predefined class. Therefore, the goal is the identification of the latent variables (i.e., PLS components) that both capture variance and attain correlation [41,42]. The evaluation of the developed discrimination models was performed by means of the venetian blinds cross-validation technique, having the number of data splits set to ten. Therefore, the performance metrics illustrating the classification ability of the honey differentiation are the results of an unbiased evaluation methodology. In the framework of the present study, the number of latent variables to keep for the construction of the PLS-DA models was chosen with the aim of minimizing the cross-validation classification error average.

The identification of the features having the highest classification power was achieved by applying a feature selection tool that relies on a comparison of the root-mean-square error of cross-validation values (RMSECV) of PLS models built on different groups of variables. Firstly, from the entire set of attributes, the ones having the lowest Variable Importance in Projection (VIP) and selectivity ratio (SR) values were eliminated from the input space; furthermore, the same procedure was followed until the performance of the PLS-DA model did not continue to improve [43].

The pretreatment technique that was used in the present work was autoscale. Through autoscale, the original matrix corresponding to the experimental data set is transformed so that each column has an average of zero and a standard deviation of one.

All of the previously mentioned data processing approaches (i.e., the application of preprocessing methods, the construction of the PLS-DA models, the selection of the most relevant variables for the development of the honey classification models) were conducted under the SOLO 8.9.1 (2021) software (Eigenvector Research, Manson, WA, USA).

## 5. Conclusions

This study proposes a new approach for the development of prediction models aiming to classify Romanian honey according to its harvesting year, botanical origin and geographical provenance by associating ATR-FTIR spectral data with analytical methods (i.e., PLS-DA). By applying, as a preprocessing step, the autoscale method, it was observed that the models gave a better classification. Dimensionality reduction through the application of the PLS-DA supervised method led, in most cases, to a drastic improvement in the classification rate. Thus, for the model developed for the harvest recognition, 105 out of 109 honey samples were correctly attributed, leading to true positive rates of 95% and 97% for the harvesting years 2020 and 2021, respectively. The markers based on which the best classification was achieved revealed that monosaccharaides and organic acids (e.g., fumaric acid) are important characteristics during the storage of honey.

The developed model for botanical origin classification presented a correct classification rate of 84% when four honey varieties were simultaneously classified. The markers selected for this classification can be explained by the differences in polyphenol concentration, water content and the interaction with the carbohydrates and proteins, whose concentrations depend on the floral origin.

The developed model for the geographical provenance allowed the correct classification of 91% of the Transylvania samples and 85% of those produced in other regions. Thus, the overall accuracy of the model developed for geographical origin recognition was 88% in the cross-validation evaluation technique.

The employment in the model construction of an extended honey sample set having a more balanced distribution could lead to the development of more robust recognition models. This subject represents a future work direction.

## Figures and Tables

**Figure 1 ijms-23-09977-f001:**
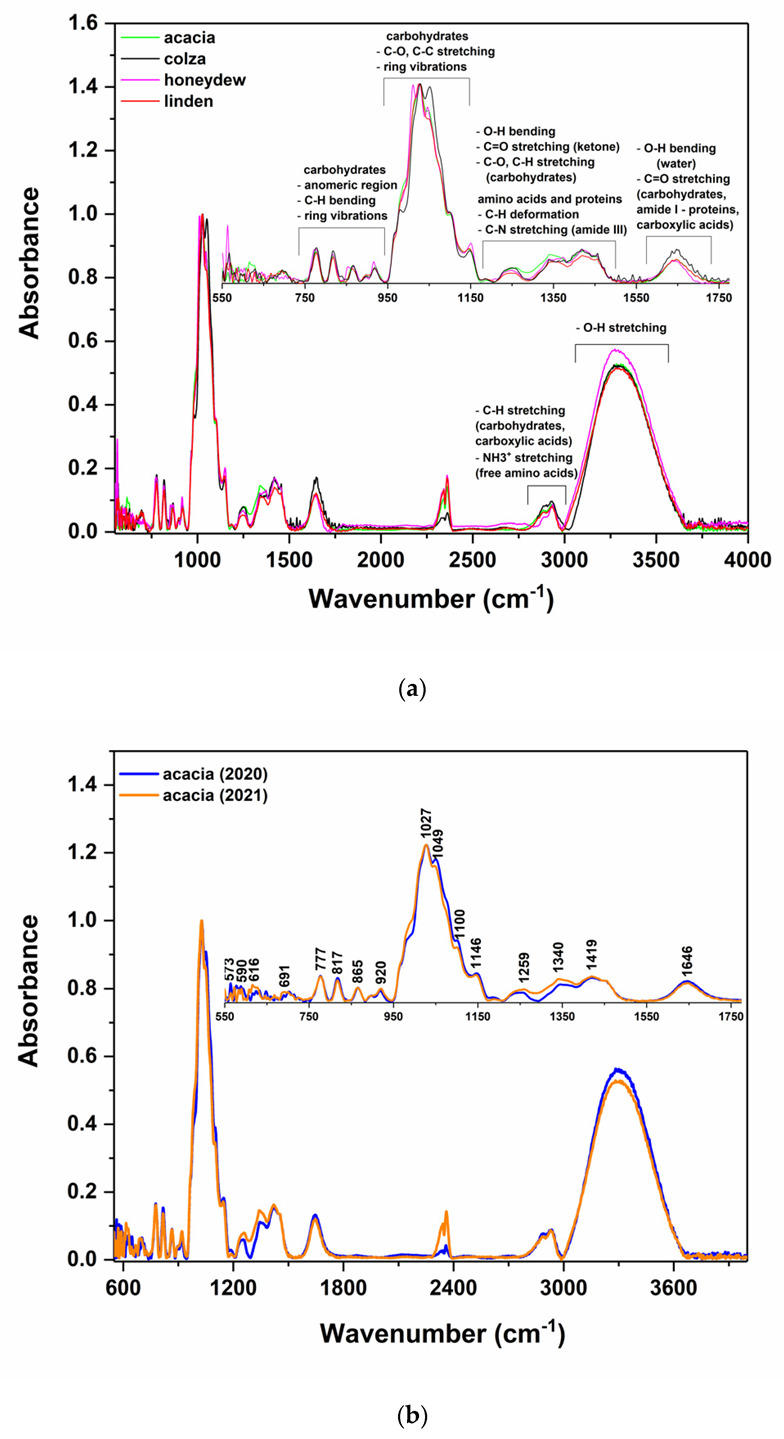
ATR-FTIR experimental spectra of (**a**) honey varieties, (**b**) different harvesting years of acacia honey.

**Figure 2 ijms-23-09977-f002:**
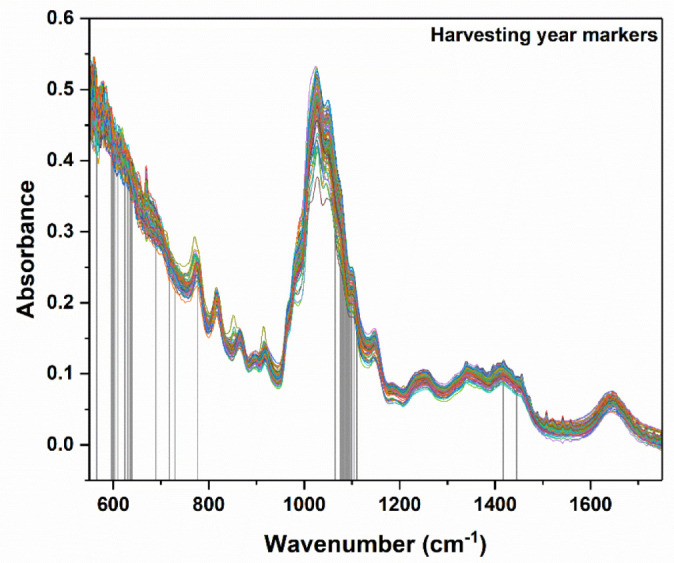
Plotted spectra of all investigated samples (for their differentiation distinct colors were used). The vertical lines marked the spectral points having high differentiation power for the harvesting year discrimination of the investigated honey samples.

**Figure 3 ijms-23-09977-f003:**
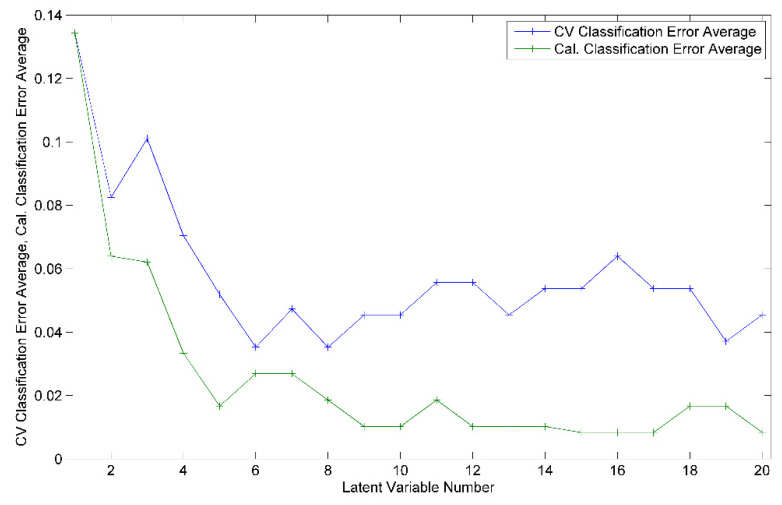
Cross-validation (CV) and calibration (Cal.) classification error average as function of the number of latent variables used for the construction of the PLS-DA harvesting year differentiation model.

**Figure 4 ijms-23-09977-f004:**
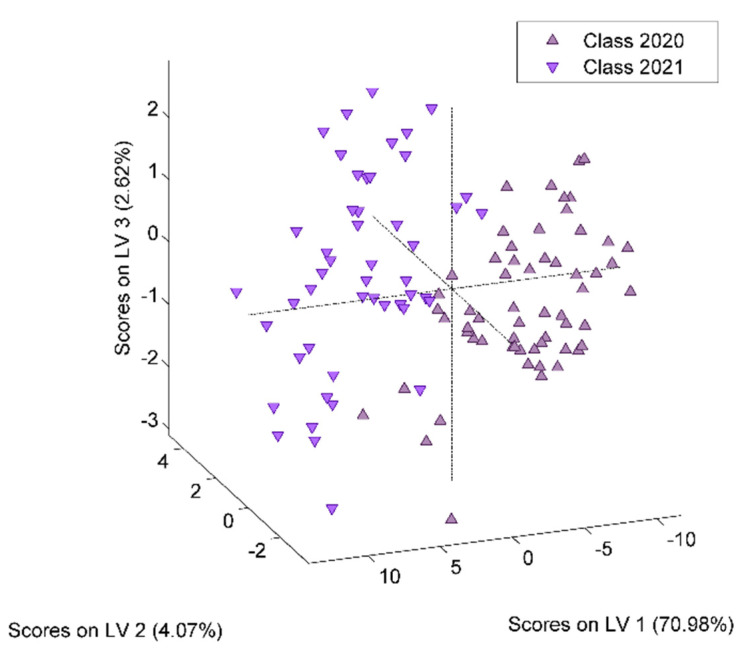
Score plot corresponding to the developed PLS-DA model for the harvesting year differentiation of the honey samples.

**Figure 5 ijms-23-09977-f005:**
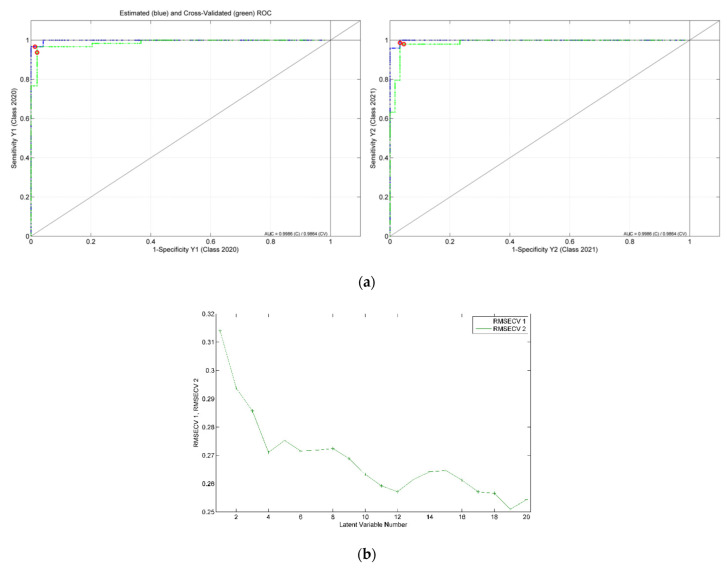
(**a**) ROC curves containing estimated (blue), cross-validated (green) values and model threshold (red circle), and (**b**) overlapped RMSECV plots for 2020 (RMSECV 1—blue) and 2021 (RMSECV 2—green) groups associated with the PLS-DA model developed for harvesting year differentiation.

**Figure 6 ijms-23-09977-f006:**
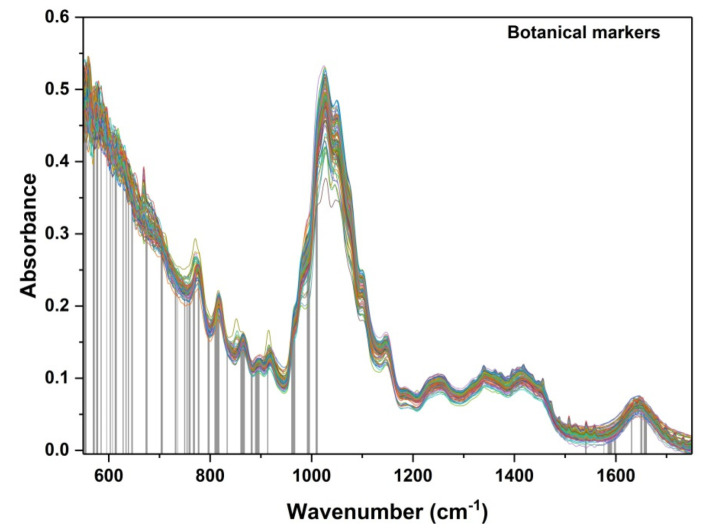
Plotted spectra of all investigated samples (for their differentiation distinct colors were used). The vertical lines marked the spectral points having high differentiation power for the botanical discrimination of the investigated honey samples.

**Figure 7 ijms-23-09977-f007:**
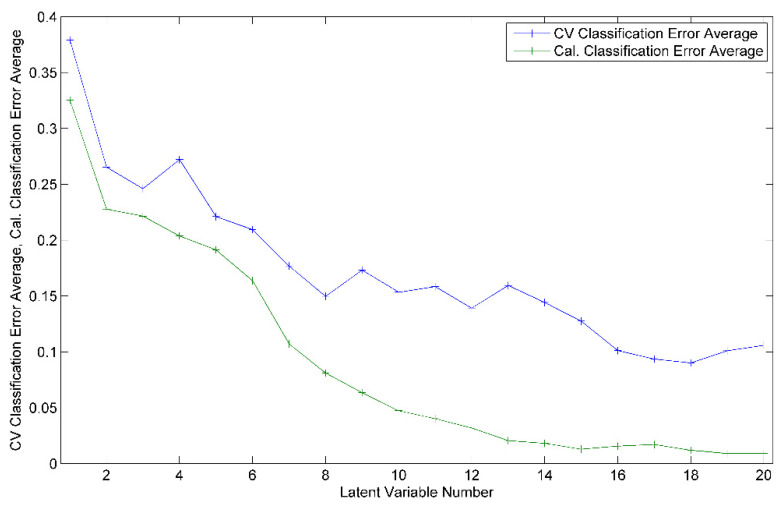
Cross-validation (CV) and calibration (Cal.) classification error average as function of the number of latent variables used for the PLS-DA model aiming at the botanical differentiation of the honey samples.

**Figure 8 ijms-23-09977-f008:**
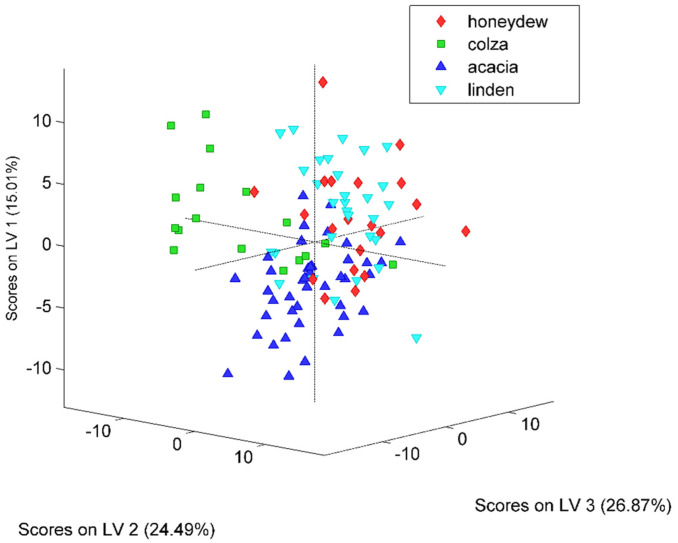
Score plot (LV1 vs. LV2 vs. LV3) associated with the PLS-DA model developed for the simultaneous differentiation of the four botanical honey classes (i.e., honeydew, colza, acacia and linden).

**Figure 9 ijms-23-09977-f009:**
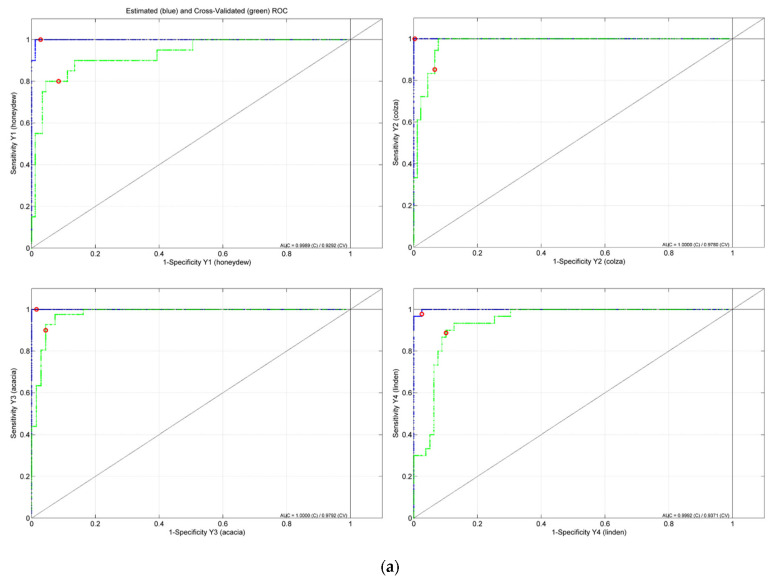
(**a**) ROC curves containing estimated (blue), cross-validated (green) values and model threshold (red circle) and (**b**) RMSECV for each of the botanical honey classes (i.e., honeydew—1, colza—2, acacia—3 and linden—4) in PLS-DA model.

**Figure 10 ijms-23-09977-f010:**
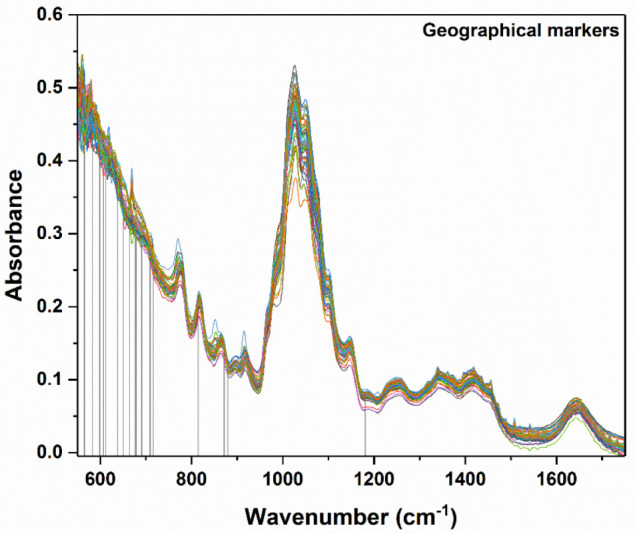
Plotted spectra of all investigated samples (for their differentiation distinct colors were used). The vertical lines marked the spectral points having high differentiation power for the geographical discrimination of the investigated honey samples.

**Figure 11 ijms-23-09977-f011:**
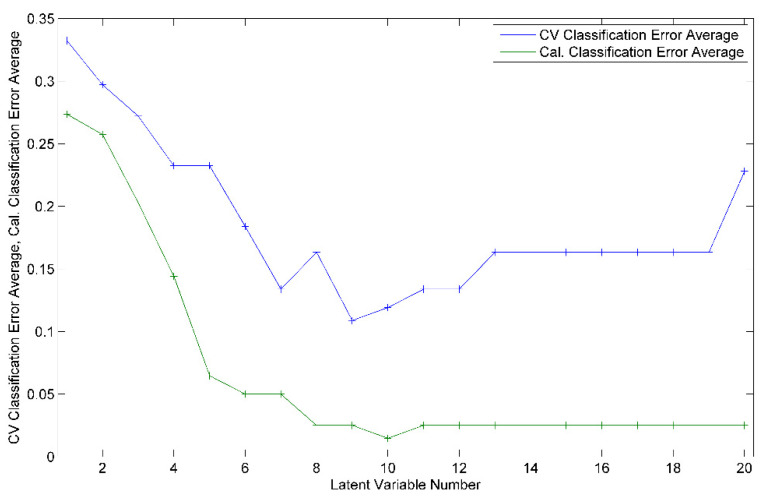
Cross-validation (CV) and calibration (Cal.) classification error average as function of the number of latent variables used to construct the PLS-DA model for the geographical differentiation of honey.

**Figure 12 ijms-23-09977-f012:**
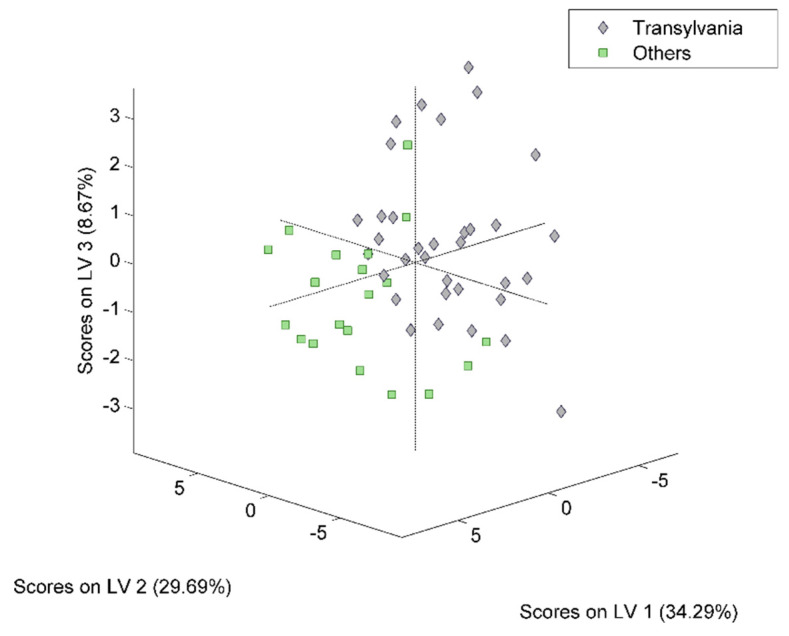
PLS-DA score plot corresponding to the geographical differentiation of honey.

**Figure 13 ijms-23-09977-f013:**
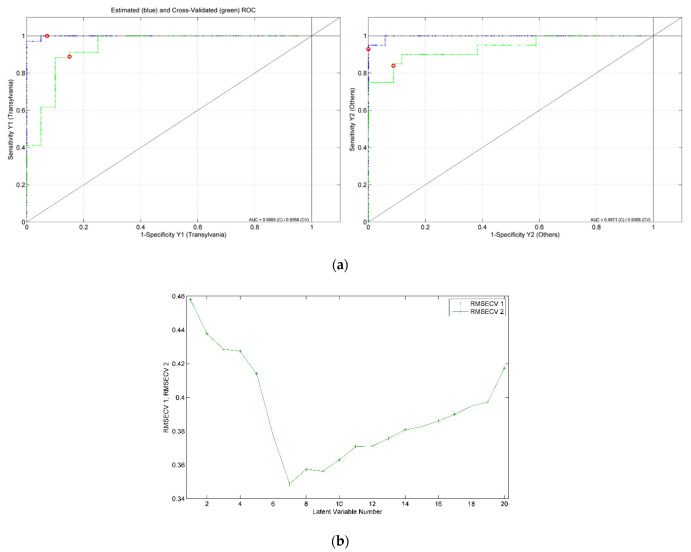
(**a**) ROC curves containing estimated (blue), cross-validated (green) values and model threshold (red circle), and (**b**) overlapped RMSECV plots for Transylvania (RMSECV 1—blue) and Others (RMSECV 2—green) classes associated with the PLS-DA model developed.

**Table 1 ijms-23-09977-t001:** The honey botanical origin, year of harvest and geographical distribution of the authentic honey samples.

Botanical Origin	Sample Number	Harvesting Year		Geographical Origin	
		2020	2021	Transylvania	Others
Acacia	41	20	21	14	6
Linden	30	19	11	7	6
Colza	18	8	10	5	4
Honeydew	20	13	7	8	4
Total	109	60	49	34	20

## Data Availability

The data presented in this study are available on request from the corresponding author.

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
