# Peer review of "The Development of Honey Recognition Models Based on the Association between ATR-IR Spectroscopy and Advanced Statistical Tools"

_ijms, 2022, doi:10.3390/ijms23179977_

Round 1

Reviewer 1 Report

Please remove the text supplied by the instructions to authors the rest of the publication is very good

Author Response

Thank you for your careful reading of our manuscript and for the suggested modifications meant to improve our manuscript!

Reviewer 1

            Please remove the text supplied by the instructions to authors the rest of the publication is very good.

            The modification was performed. Thank you!

Reviewer 2 Report

This is an interesting manuscript dealing with the authentication of honey samples based on both botanical and geographical production regions, but also on the harvesting year, which is something to highlight as commented by the authors, by employing ATR-IR spectroscopy and PLS-DA chemometrics. The manuscript is interesting, and the obtained results are good. The authors need to improve the writing, as there are many errors and sentences that are difficult to follow (there are some lines that belong to the manuscript template guidelines that were not removed).

In any case, in my opinion, the manuscript requires major revisions prior to publication in International Journal of Molecular Sciences.

First, the authors focus their research on the different classification results when using the raw ATR-IR spectroscopy fingerprinting data without and after applying some preprocessing steps, observing that the best results were always obtained when a preprocessing step was performed and when only the relevant markers were used for the development of the models. And this is expected as it has been widely reported in the literature when spectroscopy fingerprinting methodologies are employed. In my opinion, the authors need to focus only on the best proposed PLS-DA models for each classification/authentication study (harvesting year, botanical, and geographical origin), explaining how the model was built (preprocessing employed and spectral region markers used). I will recommend also to focus on the PLS-DA results, and to show in the manuscript the PLS-DA score plots and classification data (sensitivity, specificity, and cross-validation errors) for each authentication/classification study performed. This will help the reader to visually see how the samples are clustered. Also, it will be good, in some cases, to comment on the number of samples available, as for some groups the number is very limited and this can hinder, as expected, the classification rates, although in my opinion the obtained results are quite acceptable considering the authentication problem addressed.

The description of markers is well done and correctly referenced.

Other minor points:

Line 9-12: Please, revise this first sentence of the abstract, as it is not grammatically correct. Rewrite as it seems that some verbs are missing.

Line 14: “…leading to a true positive rate of 95%...” for what? Botanical origin? Geographical origin? Difficult to follow…

Line 23-24: This is not completely accurate. Honey is made from the nectar of plants, from secretions of plants, or from excretions from plant sucking insects.

Line 56: Correct the citing of the reference: “… and co-workers [10].”

Line 95, Figure 1. Please, indicate in the figure that the first one if (a) and the second one if (b). This labeling is missing.

Line 99 and line 257: Please remove “:” after “such as”. It is not required.

Lines 104-106: Remove these three lines, they belong to the manuscript template guidelines.

Lines 133-135: Rewrite. Again, it is difficult to follow. I recommend: “… from the 2020 and 2021 harvesting year, respectively, were correctly predicted…”

Lines 140-142: The same situation as the one previously commented. The term “respectively” is not correctly employed, making it difficult to understand the sentence.

Line 200: Correct cm-1 by cm-1. This error is present in other parts of the manuscript. Please, revise it carefully.

Lines 217: Correct the citing of the reference: “… (Figure 1a) [24].”

Author Response

Thank you for your careful reading of our manuscript and for the suggested modifications meant to improve our manuscript! All requested changes were addressed in the revised form of our manuscript and are hereafter detailed.

The authors need to improve the writing, as there are many errors and sentences that are difficult to follow (there are some lines that belong to the manuscript template guidelines that were not removed).

            The manuscript was improved in terms of spelling and typos and the template guidelines were deleted.

            In my opinion, the authors need to focus only on the best proposed PLS-DA models for each classification/authentication study (harvesting year, botanical, and geographical origin), explaining how the model was built (preprocessing employed and spectral region markers used).

            The manuscript was modified to present the approach that consist in the association among i) the selection of meaningful spectral range followed by ii) a pre-processing treatment (i.e. autoscale) and finally through iii) a model-based feature selection approach (i.e. based on PLS regression). Only the best PLS-DA model was compared with other methods proposed in literature in order to prove the effectiveness of our approach.

            I will recommend also to focus on the PLS-DA results, and to show in the manuscript the PLS-DA score plots and classification data (sensitivity, specificity, and cross-validation errors) for each authentication/classification study performed. This will help the reader to visually see how the samples are clustered.

            For each of the developed PLS-DA models (i.e. harvesting year, botanical and geographical differentiation), plots illustrating (i) the PLS-DA scores and (ii) the ROC curves (which present the sensitivity against 1-specificity for a given classification) were included in the manuscript according to reviewer’s observation.

            Also, it will be good, in some cases, to comment on the number of samples available, as for some groups the number is very limited and this can hinder, as expected, the classification rates, although in my opinion the obtained results are quite acceptable considering the authentication problem addressed.

            A comment regarding the extended honey sample set in order to obtain more robust recognition models was performed.

Line 9-12: Please, revise this first sentence of the abstract, as it is not grammatically correct. Rewrite, as it seems that some verbs are missing.

            The sentence was modified.

Line 14: “…leading to a true positive rate of 95%...” for what? Botanical origin? Geographical origin? Difficult to follow…

            The paragraph was modified.

Line 23-24: This is not completely accurate. Honey is made from the nectar of plants, from secretions of plants, or from excretions from plant sucking insects.

            The modification was performed according to the observation.

Line 56: Correct the citing of the reference: “… and co-workers [10].”

            The correction was performed.

Line 95, Figure 1. Please, indicate in the figure that the first one if (a) and the second one if (b). This labeling is missing.

            The figures were modified according to the observation.

Line 99 and line 257: Please remove “:” after “such as”. It is not required.

            The removal was performed.

Lines 104-106: Remove these three lines, they belong to the manuscript template guidelines.

            The lines were removed.

Lines 133-135: Rewrite. Again, it is difficult to follow. I recommend: “… from the 2020 and 2021 harvesting year, respectively, were correctly predicted…”

            The sentence was modified.

Lines 140-142: The same situation as the one previously commented. The term “respectively” is not correctly employed, making it difficult to understand the sentence.

            The sentence was rewritten.

Line 200: Correct cm-1 by cm-1. This error is present in other parts of the manuscript. Please, revise it carefully.

            The correction was applied.

Lines 217: Correct the citing of the reference: “… (Figure 1a) [24].”

            The correction was applied.

Reviewer 3 Report

The paper submitted by David et al. describes the differentiation of honey samples according to the harvest year, botanical and geographical origin on the basis of FTIR ATR spectra. Certainly, the topic may be of interest to a specific community, but improving a few points of the manuscript is a must. The topic of honey analysis using the IR technique has been widely researched, but the number of cited works is relatively small. The innovation of the research should be emphasized.

PLS-DA is a  standard method for classifying samples. For full characterization of the developed classifiers the ROC and RMSECV plots should be added. No data on the number of the LVs used for model construction before and after pretreatments.

Can PLS scores of the final models separate samples of different origin? It would be valuable to compare in the share of subsequent factors in the description of the total variability of the system before and after selection of variables; table with parameters is expected.

It is unclear if PLS-DA models were constructed on baseline corrected spectra or not. In Fig.1  shows spectra after correction, while Figs. 2-4 show uncorrected spectra. A significant number of selected variables are found in the in the low wavenumber region, for which there is considerable variability.

In the lines 263-267 instructions to the authors are cited.

Author Response

Thank you for your careful reading of our manuscript and for the suggested modifications meant to improve our manuscript! All requested changes were addressed in the revised form of our manuscript and are hereafter detailed.

The topic of honey analysis using the IR technique has been widely researched, but the number of cited works is relatively small. The innovation of the research should be emphasized.

            The literature paragraph was modified and more studies were included and discussed. The innovation of the research was emphasized.

            PLS-DA is a standard method for classifying samples. For full characterization of the developed classifiers the ROC and RMSECV plots should be added. No data on the number of the LVs used for model construction before and after pretreatments.

            The ROC and RMSECV plots were included in the manuscript for each of the investigated classifications. Moreover, the number of the latent variables used for the construction of the PLS-DA models was specified and justified by adding plots illustrating the Cross-validation (CV) and Calibration (Cal.) classification error average as function of the number of components.

            Can PLS scores of the final models separate samples of different origin? It would be valuable to compare in the share of subsequent factors in the description of the total variability of the system before and after selection of variables; table with parameters is expected.

            According to the observations received by Reviewer 2, the manuscript was modified by keeping in the discussion only the best obtained model and describing only the pretreatment that was used for each obtainment. Therefore, to be also more aligned with the journal’s aim, we maintained only the description of the most accurate classification models and we focused more on the chemical composition of the investigated matrix.

            It is unclear if PLS-DA models were constructed on baseline corrected spectra or not. In Fig.1 shows spectra after correction, while Figs. 2-4 show uncorrected spectra.

            The models were constructed without baseline correction. In Figure 1. a baseline correction was applied in order to better visualize the characteristic vibrations from each spectral region. Figure 2-4 illustrate the raw data (i.e. without baseline correction) that were used as input data for the statistical treatment.

            A significant number of selected variables are found in the in the low wavenumber region, for which there is considerable variability.

            It is normal to have distinct predictors for different classifications. This is because, for instance, in the case of the harvesting year classification, the differentiation is made based on monosaccharaides concentration that are an important feature for storage time of honey and also the region corresponds to M-O vibrations which offer information about the metals from soil and environmental conditions. While, in the case of the geographical discrimination, important markers can be found in the low region (i.e. 500 – 700 cm-1), being specific to M-O vibrations (Al, Cu, Mn, Pb, Fe, Li, Sr), which are characteristic features for geographical differentiation.

In the lines 263-267 instructions to the authors are cited.

            The lines were removed.

Round 2

Reviewer 2 Report

The authors correctly addressed my revision comments. The manuscript can be accepted for publication.

Reviewer 3 Report

the corrections made are satisfactory